# “*Don’t Bring Me a Dog…I’ll Just Keep It*”: Understanding Unplanned Dog Acquisitions Amongst a Sample of Dog Owners Attending Canine Health and Welfare Community Events in the United Kingdom

**DOI:** 10.3390/ani11030605

**Published:** 2021-02-25

**Authors:** Katrina E. Holland, Rebecca Mead, Rachel A. Casey, Melissa M. Upjohn, Robert M. Christley

**Affiliations:** Dogs Trust, 17 Wakley Street, London EC1V 7RQ, UK; rebecca.mead@dogstrust.org.uk (R.M.); rachel.casey@dogstrust.org.uk (R.A.C.); melissa.upjohn@dogstrust.org.uk (M.M.U.); robert.christley@dogstrust.org.uk (R.M.C.)

**Keywords:** dogs, dog acquisition, dog ownership, pets, qualitative research

## Abstract

**Simple Summary:**

Dogs are the most common companion animal in the United Kingdom; however, the pet dog acquisition process is not fully understood. It is important that stakeholders, including those working in the canine welfare sector, understand this process to enable them to provide appropriate support for dog owners across the stages of acquisition and ownership. This paper reports on qualitative findings from interviews conducted with dog owners, which sought to understand their motivations for dog acquisition. It was found that many owners had not intended to acquire a dog. In some cases, people had taken on a dog when a relative or friend became unable to care for the dog, whilst others had happened upon a dog in need. Emotional connections with the dog or a desire to help an animal in need were commonly reported motivations for keeping the dog. More research is needed to understand how common unplanned acquisitions are.

**Abstract:**

Understanding the factors that result in people becoming dog owners is key to developing messaging around responsible acquisition and providing appropriate support for prospective owners to ensure a strong dog–owner bond and optimise dog welfare. This qualitative study investigated factors that influence pet dog acquisition. Semi-structured interviews were conducted with 142 sets of dog owners/caretakers at 23 Dogs Trust community events. Interviews focused on the motivations and influences that impacted how people acquired their dogs. Transcribed interviews and notes were thematically analysed. Two acquisition types were reported, that each accounted for half of our interviewees’ experiences: planned and unplanned. Whilst planned acquisitions involved an intentional search for a dog, unplanned acquisitions occurred following an unexpected and unsought opportunity to acquire one. Unplanned acquisitions frequently involved a participant’s family or friends, people happening upon a dog in need, or dogs received as gifts. Motivations for deciding to take the dog included emotional attachments and a desire to help a vulnerable animal. Many reported making the decision to acquire the dog without hesitation and without conducting any pre-acquisition research. These findings present valuable insights for designers of interventions promoting responsible acquisition and ownership, because there is minimal opportunity to deliver messaging with these unplanned acquisitions. Additionally, these findings may guide future research to develop more complete understandings of the acquisition process. Further studies are required to understand the prevalence of unplanned acquisitions.

## 1. Introduction

Despite the popularity of pet dogs in many parts of the world, including the United Kingdom where one-quarter of households report including one or more dogs [1], the process of dog acquisition is not well understood. This oversight matters for dog welfare, because the decision-making of prospective owners may affect the quality of the human–dog relationship [2]. In addition, previous research indicates that failure to consider the time investment and financial costs involved in dog ownership could result in relinquishment [3].

Dog acquisition research has predominantly focused on exploring the human- and dog-related factors associated with four main aspects: motivations for dog ownership in general; the type of dog acquired; the individual dog acquired; and the source from which the dog is acquired. A review [4] of the current evidence summarised findings and highlighted the complexity of the multiple factors involved in the dog acquisition decision-making process. Dog acquisition decisions are influenced by both canine and human characteristics, with previous research identifying several human factors that may influence acquisition decisions. For instance, studies have identified that prior experience with dogs influences later dog ownership [5,6]. Other research has explored how socioeconomic and demographic factors, such as education attainment [5,7,8], gender [9], and age [9,10] influence dog acquisition practices and views. Regarding breed popularity, research suggests that breeds follow fluctuations typical of fashions and fads [11]. This may, unfortunately, lead some individuals to select a puppy who, when fully grown, is unsuited to the owner’s lifestyle [11]. Various canine characteristics, including the dog’s physical appearance, health, and behaviour, may influence acquisition decisions. Several studies have identified that physical appearance is a key consideration for potential owners [12,13,14,15,16]. In addition, research indicates that potential owners may value appearance over health considerations [12,17]. Other evidence suggests that the importance of the dog’s appearance may vary between owners of different breeds [16]. A dog’s behaviour can also play a role in acquisition decisions, with studies conducted within shelter environments demonstrating that the types of behaviours displayed by dogs can impact their length of stay [18]. Other studies have explored the topic of pre-purchase research, indicating that around one-fifth of prospective owners do not undertake any research before acquiring a dog [19,20].

Whilst considerable academic interest in the factors influencing dog acquisition is evident, much of the current evidence relies on data collected using survey methods that tend to restrict the degree to which participants can express their experiences. A reliance on this, largely quantitative method of data collection alone offers a limited view of the complex factors influencing the acquisition experience. Qualitative methods have been used much less frequently to explore this topic, but one exception is a 2008 study of the meanings and roles that pets play in their owners’ lives [21]. From in-depth interviews with six pet owners in the United States, it was found that two types of acquisition circumstances comprised the participants’ experiences: planned and unplanned. Planned acquisitions account for the “proactive, deliberate search for a pet” [21] p.518, and decisions about which species or individual animal to acquire were influenced by factors including housing and previous pet experiences. Unplanned acquisitions, on the other hand, occurred when an opportunity to acquire a pet was presented when the person was not seeking one. A principal theme that emerged amongst these owners was an immediate or quick emotional connection to the animal. This finding is similar to that reported by Irvine, in her study of the adoption decision process [22]. From research conducted at one animal shelter in the United States, Irvine [22] identified that adopters fell into one of three categories: (1) “planners”—who knew what kind of animal they were looking for; (2) “impartials”—who were more open-minded about the animal’s specific characteristics; and (3) “smittens”—who were drawn towards particular animals by an “irresistible pull”, sometimes without having prior plans to adopt an animal. Those described as “smittens” typically felt a connection with an animal from the outset of their adoption decision, in contrast with “planners” who tended to first identify an animal who met their search criteria.

Previous research has thus indicated that some people may acquire a dog despite not intending to. Understanding the different types of acquisition (i.e., planned or unplanned) may enable prospective owners to be better targeted with information, thus improving the quality of the human–dog relationship and reducing the risk of relinquishment. However, due to the small sample sizes of previous studies, the scale of unplanned acquisitions remains unknown. Generalisation of Mosteller’s [21] findings are further prohibited due to the study’s recruitment of participants based on professional experience with the pet industry. Building upon these previous studies, our study expands the investigation of dog acquisition behaviour into a different geographic location (the United Kingdom). The current study used semi-structured interviews with dog owners across the United Kingdom to investigate factors that influenced their experiences of obtaining dogs. These interviews were conducted as part of a wider Dogs Trust study investigating owner behaviour and motivations during dog acquisition.

## 2. Materials and Methods 

Ethical approval for this study was granted by the Dogs Trust Ethical Review Board (reference number: ERB018).

### 2.1. Participant Recruitment

Participants were recruited at Dogs Trust community events. These free events allowed owners to obtain advice on topics including diet, exercise, and enrichment. A free on-the-spot microchipping service was also offered, and qualified veterinary nurses (excluding Northern Ireland events) provided free basic health checks. Furthermore, Dogs Trust offered subsidised neutering vouchers to attending owners of dogs considered “at risk”, for example, situations in which an entire male and female lived in the same house. The locations for these events were determined in part using findings from the annual Dogs Trust Stray Dogs Survey [23]. The Stray Dogs Survey investigates the number of stray dogs handled by local authorities in the United Kingdom each year, what proportion were microchipped and what the outcomes were for the dogs (e.g., whether they were they reunited with their owner, rehomed by local authorities, put to sleep, etc.). Within areas of concern identified from the Stray Dogs Survey data, community partners’ (e.g., dog wardens and housing association staff) knowledge regarding local hotspots for dog-related issues and areas of deprivation, as well as venue availability, helped to establish event locations.

Between May and December 2019, 23 community events were attended by at least one of the authors (K.E.H., R.M., or R.M.C). The interview strategy aimed to capture experiences from owners across the United Kingdom, with events attended in London (*n* = 5), North East England (*n* = 4), North West England (*n* = 2), Northern Ireland (*n* = 4), Scotland (*n* = 3) and Wales (*n* = 5). Participants were approached whilst waiting to speak to the event staff, or after the member of staff had attended to them (and their dog). The nature and purpose of the interview was explained by the researcher and they were invited to take part on-the-spot. Participants were required to be 18 years old or above to take part. Written informed consent was obtained from all participants prior to interviews. 

### 2.2. Interview Process

Interviews explored owners’ experiences of dog acquisition, focusing on the dog(s) they had with them at the event, as well as other currently or previously owned dogs. Participants were questioned about where they obtained their dog(s) and what factors influenced their choice of dog type and source. A semi-structured interview guide (Appendix A), piloted with eligible Dogs Trust staff before use, was used to ensure that broad topics of interest were covered whilst enabling participants the freedom to articulate their experiences in their own terms.

### 2.3. Thematic Analysis

Participants had the option of whether they consented to audio recording. Where participants gave consent, interviews were audio recorded and professionally transcribed intelligent verbatim. For interviews where participants did not give consent for audio recording, or where events were too noisy to enable clear audio recordings, handwritten notes were made and subsequently digitised by the researcher who conducted the interview. Pseudonyms were created for each owner and dog. Interview transcripts and notes were imported into NVivo (v.12, QSR) and analysed using inductive thematic analysis. Themes were inductively identified from semantic and latent codes [24]. Through systematic data coding, key themes around owners’ experiences of dog acquisition emerged. 

## 3. Results

### 3.1. Participant’s Characteristics

Interviews were conducted with 142 owners or carers (or sets of owners, where a dog was accompanied by more than one co-owner). Participants were not asked to provide demographic information. However, interviewer’s notes indicated the inclusion of men and women of a range of ages. The majority of interviews lasted between 2 and 26 min in length. The mean duration was 11 min.

Across our participants, two distinct types of acquisition circumstances become apparent: planned and unplanned. Here, “unplanned” refers to cases in which owners were not actively seeking to acquire a dog when the opportunity to acquire them was presented. Each type of acquisition accounted for half of the participants’ acquisition experiences, with seventy-one participants (50%) reporting experiences we classified as unplanned acquisitions. The findings presented below refer to themes that emerged amongst those who acquired their dog in an unplanned manner.

### 3.2. How the Opportunity to Acquire the Dog Occurred

Our participants were presented with the opportunity to acquire a dog unexpectedly in a variety of ways. In three cases, the participant’s dog had an accidental litter, from which the participant kept a dog. However, most cases of unplanned acquisitions could be grouped within the following themes: those who took ownership of a friend’s or relative’s dog; those who happened upon a dog in need of a home; those who received their dog as a gift; and those who “fell in love” with a dog whilst helping a friend or relative search for one.

#### 3.2.1. Family or Friends

Family and friends were a key source of dogs that were acquired unexpectedly. Of those participants who had acquired a dog unexpectedly, 38 (53.5%) reported family or friends as the source. In some cases, major life-changing events affecting family or friends prompted a sudden need to find the dog a new home. Common events included illness, death, pregnancy or the arrival of a baby, and housing changes. Other less frequently reported events included divorce and prison sentences. On some occasions, people were asked, by the previous owner, to take the dog. Some were happy and willing to help out, as this owner explained:

“We’ve got the dog that I’ve got because my friend went into hospital and couldn’t look after him. She said, ‘can you have the dog?’ (I said) ‘yeah fine, no problem’.”(Jessica)

Others saw an opportunity to help and offered to take the dog, rather than being asked. One owner volunteered to take a dog, who they were particularly fond of, from their friend who was moving overseas:

“He got a job out the country and couldn’t take all of his dogs with him and I asked him, could I take Lottie.”(Melanie)

However, a different experience was reported by one participant who “inherited” their dog following the death of a relative. Recalling their acquisition, this owner expressed that they lacked an active choice in the matter:

“I moved in with my nan when she was unwell, and then she passed away so I kind of just got given the dog.”(Stacey)

Owners’ family and friends were also the source of unplanned acquisitions for reasons that were less life-changing, for instance where a friend or relative had become bored of their dog. In one case, it was commented that the previous owners had originally acquired the dog for a child, who subsequently grew bored of the dog’s novelty. This participant explained that following a period of them providing temporary care for the dog, the dog’s previous owner saw an opportunity for him to experience an improved quality of life in the care of the new owner:

“It was my ex-daughter-in-law. She got it for the children. But she’s working, and she had a dog walker just come in to take him out. I took him for their holidays. She thought it was a shame for him to come back to the house and being shut in.” (Carol)

In several cases, participants’ dogs had been acquired from friends or relatives whose dog had had a—sometimes “accidental”—litter. One participant explained how their neighbour had had an accidental litter of three puppies. Although their neighbour had planned to keep all three, after a few months they decided they could not cope with them so wanted to rehome one. Having seen the puppies grow up from birth, the participant explained that they felt compelled to take a dog. In another case, a participant had not intended to acquire another dog after their sadness of losing a previous dog, but spoke of “falling in love” when they saw their daughter’s dog’s puppy.

#### 3.2.2. Happening Upon a Dog in Need

A commonly reported way in which the opportunity to acquire a dog in an unplanned manner arose was cases in which the owner, or someone they knew, came across a dog whose welfare needed protecting due to a range of unfortunate circumstances. Happening upon a dog in need was reported by 21 participants, accounting for 29.6% of those who acquired a dog unexpectedly. These included instances of becoming aware of dogs who were being mistreated by someone in their local community and encountering abandoned dogs, including a dog tied to railings outside the acquirer’s home and another tied to a tree in a park. For instance, one participant described how they had happened upon an abandoned dog outside their home:

“And then we came back from going down to the beach one day, and…someone had abandoned a puppy.” (Jessica)

Another participant recalled their surprise at finding a puppy in the boot (trunk) of an unattended vehicle:

“I kept hearing this squealing like, whimpering and I jimmied the boot up and she was in it.”(David)

For another participant, the opportunity to acquire a dog in need of a home occurred unexpectedly when their relative required help to care for an abandoned dog. Whilst the participant had initially intended to provide only temporary care for the dog, this soon became a permanent acquisition because they quickly formed an attachment to the dog:

Participant: “So, my sister-in-law used to work in a veterinary practice, and he was just brought in as a rescue, just been dumped somewhere, and they took him home, but they just, they’ve got three small kids and they’ve not got a huge place.”Interviewer: “Okay, so you took him in for a bit?”Participant: “Yeah, and then we fell in love with him, and then took him on [laughs].”Interviewer: “But you weren’t looking for a dog at all?”Participant: *“*No*.”*(Sarah)

Furthermore, two participants had been contacted with a proposition about a dog in need of a home from a breeder from whom they had acquired one in the past.

#### 3.2.3. Dogs Received as Gifts

Some participants had received their dog as a gift about which they had not been consulted (9.8%, *n* = 7). Typically, dogs were gifted between family members and were given to mark an occasion, such as a birthday, Christmas, or a retirement. This unexpected gift prompted surprise in some recipients:

Participant: “Well, the story goes, I was retiring, and my wife bought the dog, unknown to me.”Interviewer: “She bought the dog, unknown to you?”Participant: “Well she thought it would be a good retirement present. I’ve never had a dog before, so it was all new to me…Well, as soon as I found out I was a bit shocked.”(Christopher)

#### 3.2.4. Happening Upon Their Dog Whilst Helping a Relative to Find a Dog

Several owners, or their partners, “came across” their dog whilst helping a relative to look for a dog to acquire (4.2%, *n* = 3). In the process of supporting a relative, by accompanying them to meet litters of puppies, they fell in love with a particular dog, as this participant explained:

“My husband went to help his sister look for a dog…he’s had Springers, so she said, ‘will you come and help me and have a look at this littler Sprocker?’ (Springer Spaniel X Cocker Spaniel)…So off he toddled to help her, one Saturday morning, she was looking at some Sprockers and some Labs. So, when he’d finished helping her with looking at the Sprockers, off she toddled and about an hour later, he got a picture on WhatsApp of a little baby Lab, (his sister had) chosen one. And so, then he turned round to me and went, ‘that little Sprocker needs a home now, do you want to go and see it?’”(Dawn)

### 3.3. Deciding to Acquire the Dog

Across the cases of unplanned acquisitions, four themes emerged that help to explain why owners decided to take on the dog when they were presented with the opportunity, despite not seeking one at the time. These themes were: prior relationship with the dog; quick emotional attachments; the dog’s vulnerability; and the dog’s physical appearance.

#### 3.3.1. Prior Relationship with the Dog

Some owners were already familiar with the dog that they acquired. This was common where the dog had previously been owned by family or friends. In these instances, owners had often known the dog for much, or all, of their life, sometimes also having occasionally cared for the dog prior to acquisition and thus had typically already developed a positive relationship with the dog. For example, having seen their (subsequently acquired) dog grow up with their neighbour, one participant felt that they “couldn’t not take him” when they became aware that the dog needed a new home. Another participant explains how their dog had been “in the family” for many years prior to acquiring her when their mother-in-law moved into a smaller house:

“We have had her in the family for six years…So they were around all the time, obviously, visiting and things, and now she’s ours.”(Faye)

#### 3.3.2. Quick Emotional Attachments

An immediate connection to the dog was frequently noted, as participants explained their decision to acquire the dog. Quick emotional attachments were particularly prevalent amongst owners who did not have a relationship with the dog prior to acquisition. Many spoke about “falling in love” with the dog. These intense emotional attachments appeared to take precedence over any knowledge or intentions the owner might have had about whether acquiring the dog was the right decision for them. For example, one participant explained how they had accompanied their relative, who was looking to obtain a dog, to visit a litter of French Bulldog puppies. Despite the participant having reservations about brachycephalic breed-related health issues, commenting that they were keen to not fall into the “brachy trap”, as they put it, they immediately fell in love with a puppy who they decided to purchase straight away.

Several owners had taken their dog in on a temporary basis at first, often to protect the dog’s welfare. Although they did not intend to keep the dog permanently, the rapid development of an emotional attachment to the dog often encouraged them to keep the dog:

“I met him at work and fell in love. And I was like, ‘Oh my God, who’s this?’ She (the participant’s colleague) said he needed to be adopted ‘cos she couldn’t keep him. The shelter that she got him from originally didn’t really get back to her very much. So I was like, ‘I’ll have him for a couple of weeks, see how it goes’. And then, I was like, ‘I can’t give him back, I love him.’”(Harriet)

In addition to feelings owners developed towards the dogs, participants described attachment in reciprocal terms, suggesting the importance of the dog–human bond in some owners’ decision-making. For instance, a participant who was not initially sure whether they would keep the dog they found tied to railings outside their house explained how they were motivated to keep her as she “took to” them. Another participant, who obtained their dog from a friend, described the bond they felt with their dog that had developed over several years prior to acquisition. They noted the close physical contact they experienced with the dog:

“She paid me a lot of attention. Every time I went ‘round his (their friend’s) house she wanted to sit on my knee and she was always climbing up and over me. It was probably because I was the only one paying her any attention but she seemed to just click with me, she liked me, I liked her and when I had the opportunity, I just pounced and (asked) ‘Can I have her, please?’”(Melanie)

#### 3.3.3. The Dog’s Vulnerability

Many participants emphasised their dog’s vulnerability at the time of acquisition, linking this with their decision to acquire the dog. In some cases, such vulnerability was associated with a health problem. One participant connected their dog’s deafness to their decision to acquire the dog:

“She was just really, really cute and she was deaf as well. So, I was like, okay I have to have her.” (Ellie)

In another case, a participant who had come across an abandoned dog commented on the poor condition in which they found the dog to be:

“(He was) covered in fleas. Oh in a terrible state… So we had him.” (Jessica)

For some others, the dog’s vulnerability was linked to their unfortunate treatment at the hands of their previous owners. Again, this was often reported as having an important impact on the decision to take the dog on:

“I knew she was badly treated and it would just break my heart, so there wasn’t a thought in it really. It was like, right okay.”(Sue)

Concerns about alternative options for the dog if they did not give them a home were also expressed by some participants. Sometimes this included a desire for the dog to avoid ending up in a shelter:

“She (the dog’s previous owner) was like, ‘I need to give it to somebody or I’m just gonna give her to a shelter.’ I was like, ‘no, I’ll take her.’”(Ben)

Others commented that they were worried about whose hands the dog might end up in, with one owner reporting that she was concerned the dog could end up being used as bait for dog fighting. Another concern was that the dog may be at risk of being put to sleep if they did not provide them with a home:

Participant: “So, it was, what would you say, it was spontaneous. It wasn’t that well thought out. We didn’t do all the research about what breed of dog to get, or anything. It was just, this dog—The woman actually said to my friend, ‘I’m going to have her put down if I can’t find a home for her because she’s so neurotic’.”Interviewer: “And did they say that to you?”Participant: “Yeah. So we thought ‘Oh no! We couldn’t possibly let you do that’.”(Marie)

The speed of the decision to acquire the dog was often associated with the participant’s assessment of the dog’s current situation and potential alternative outcomes should they not take the dog. As in the above account, many participants who “rescued” their dog from unfortunate circumstances reported that they made the decision to do so immediately or very soon after the opportunity was presented to them. Often this decision was made instantly without taking pause to consider the longer-term implications of such a decision:

“I was like, ‘no, I’ll take her, I don’t even need to think about it, just tell me when.’”(Ben)

Decisions to acquire an unplanned dog were often influenced by various factors. One participant, whose sister’s dog had had an accidental litter, explained that her sister was particularly fond of one of the puppies, Molly, due to her physical resemblance to her mother. Describing the circumstances around this acquisition, the participant cited several factors, including the tragic outcomes of some of Molly’s littermates, her sister’s concerns about finding Molly a “good home” and desire to maintain contact with Molly:

Participant: “My sister really wanted a good home for Molly and to keep her close so that she can still see her. And some of the dogs didn’t really get good homes. And she had to rescue some of them. Some of them died.”Interviewer: “So were you looking for a dog?”Participant: “Not at the time, no. But because I knew that Molly needed a good home, I was just like ‘yeah, I can give her a good home.’… It was a decision that I just took on the spur of the moment.”(Joanne)

Some participants who were presented with a dog which they felt was in need emphasised that overall, the individual dog’s characteristics (e.g., breed or appearance) was not important to their decision to acquire the dog. Rather, they would have offered a home to any dog that was presented to them in need:

“Don’t bring me a dog and show me it, a puppy, I’ll just keep it.” (Sue)

“I mean, it could have been any dog and I’d have took it (sic).” (Carla)

#### 3.3.4. The Dog’s Physical Appearance

While some participants suggested that they would have taken on any dog in need, regardless of factors including appearance, a few people associated the dog’s physical appearance with their decision to acquire the dog. For these participants, the dog’s appearance was associated with their emotional response towards the opportunity to acquire the dog:

“Well you go soft, don’t you? He was so cute, I’ve showed you the picture.”(Dawn)

Interviewer: “Were you looking for a dog at the time when you saw it (the advert)?”Participant: “Not really. We were open to the idea but we weren’t really looking and then my wife suddenly saw photos of them.”Interviewer: “She just saw the photos?”Participant: “Yes and we thought they looked adorable, we’ll go and have a look at them…Once you see them, you seem to know.”Interviewer: “What was it about them when you saw them?”Participant: “It was just really their faces and it’s just puppy love.”(Harry)

In addition to comments about the dog’s attractiveness in general, several participants cited their attraction to the dog’s coat colour, and one mentioned their attraction toward specific features of their dog:

“She was brought in (to the veterinary practice where the participant worked) as a stray…I wanted her then because she was so cute…the way her teeth and stuff come out and the big puggy brown eyes.” (Ellie)

A couple of participants referred to the small size of their dog. A small dog was considered more manageable by some, but in another case was suggestive of the dog’s vulnerability:

“I phoned my girlfriend at first and I said to her, “she’s beautiful.” She was only like a wee ball, you know.”(David)

### 3.4. Inconsistency Between Plans for a Future Dog and the Actual (Unplanned) Acquisition

Several participants explained that although they had not intended to acquire a dog at the time at which they did, they did have plans to obtain a dog at some point in the future when their lifestyle could accommodate a dog. For instance, one participant was planning to hold off until their retirement. However, becoming aware of a dog in immediate need of a home led some to “take the plunge” into dog ownership sooner than they had anticipated. There was, therefore, a mismatch between the timing of their imagined future acquisition and the timing of the actual, unplanned, acquisition:

“When I saw him, I was like, okay, he can’t go to like loads more homes. So I was willing to speed it up a bit.” (Harriet)

The sense of urgency and/or emotional attachment that was a feature of many of our participants’ experiences also led some to acquire dogs with characteristics (e.g., age or breed) that were inconsistent with the “future dog” they had in mind. For instance, while the above-mentioned participant had intended to acquire a puppy in the future, they found themselves taking ownership of a slightly older (seven-month-old) rescue dog due to a chance meeting with the dog via a colleague: 

“I met him at work and fell in love… (We had planned to get a dog) a bit later on. But he—we were going to get a puppy not a rescue, because they’re easier.”(Harriet)

Other mismatches between an imagined dog and the dog which was actually acquired concerned the dog’s breed. This is illustrated in the earlier example of a participant who had strong feelings against the purchase of brachycephalic dogs and did not want to fall into the “brachy trap”, although nevertheless made an immediate decision to acquire a French Bulldog puppy when feeling a connection with the animal. Another participant who cited their emotional connection to the dog as a motivator for their acquisition similarly recognised that the type of dog they acquired—a Dachshund—conflicted with their ethical views concerning purebred dogs:

Participant: “I always thought if I had a dog, I would get a cross breed or mongrel.”Interviewer: “Why was that?”Participant: “I don’t necessarily always agree with pedigrees because some of them have so many problems, so many bad health problems, which are bred into them. One of my friends, he had British Bulldogs and, as lovely dogs as they are, I’ve seen all their health problems and that’s why I quite often don’t agree with pedigrees. And, quite often, they’re inbred too, some of them. But I just liked her (the dog she acquired).”(Melanie)

Despite not resembling the kind of dog they had in mind for a future acquisition, most participants reported having a largely positive relationship with their dog who they “would not be without”:

“And in the end, now, we’ve got him and would never change him. I absolutely love him.”(Harriet)

However, some participants encountered challenges or undesirable dog behaviour. For one participant, who had acquired their dog seven years earlier through a mutual friend who was urgently seeking a new home for the dog due to behavioural issues, their dog had not met their ideals: 

“It was a long time for her to be confident. But she’s never really—she still won’t go to my husband…Not unless he lies on the floor with his arm under his bottom, you know, like no threats…So, she is lovely but she’s not the fun dog that we would have chosen if we’d been more thoughtful about it.” (Marie)

### 3.5. Maintaining Contact with Former Owners

Where dogs were acquired from a prior owner (e.g., when a previous owner sought a new home for their dog due to changes in their living circumstances or work routine) it was common for the former owner (often a friend or relative) to still see the dog, either often or occasionally:

“She sees him a lot. I mean, she didn’t want to give him up but she just couldn’t (keep him). She lives in a flat.” (Harriet)

“They just live around the corner so they’re round all the time.”(Faye)

Even when the previous owner of a dog did not live nearby and therefore could not visit, participants updated them on the dog by sending photos:

“He’s quite happy. To this day, I still send him pictures of her on Facebook and things like that, so he does still keep in contact with her…And the odd time he’s back in (the country) he’ll come round and visit and she still remembers him.”(Melanie)

“Sometimes we connect them, send them pictures and all that.”(Jane)

## 4. Discussion

This study used semi-structured interviews to explore how and why people acquire dogs. We found that a significant proportion of the dog owners interviewed had not intended to acquire a dog at the time of their acquisition. These results cast a new light on the dog acquisition process, with implications for the design of future research. With a view to improving the validity of future surveys, we suggest that researchers should design questions and response options that allow unplanned acquisitions to be incorporated.

Although this paper focuses on unplanned dog acquisition, some of the factors found to drive unplanned acquisition decisions amongst our participants have been identified in previous research exploring dog acquisition more broadly. Notably, our finding regarding the role of a dog’s physical appearance as a driver for unplanned acquisitions is supported by previous evidence. For example, among Finnish dog owners, physical appearance was found to be of greater importance in affecting their decision to acquire a specified breed when compared to the risk of serious breed-associated genetic diseases [17]. In another study among people adopting dogs from animal shelters, a greater percentage of adopters reported the dog’s physical appearance as important (75%) when compared to the proportion who rated the dog’s health as important (49%) [12]. Our findings provide further evidence for the role that appearance can play in driving acquisition decisions, even when the acquisition was not planned.

It is clear from our study that a dog’s perceived vulnerability may be an important motivating factor influencing people to acquire a dog, despite not having a prior intention to do so. This finding might be understood in relation to previous research that identifies evidence for the innate human instinct to care for a dog. A specific set of infantile features, including a large head, round face and large, low-lying eyes, were described in classical ethology as evoking a nurturing response from human caregivers [25]. This human attraction to infant-like stimuli has been generalised to cats and dogs [26], with evidence suggesting that people demonstrate the same degree of empathy for a baby as for a puppy [27]. The ability of puppies to eclipse rational human thought, due to the “cute response” is suggested to be an important motivation to acquire a puppy [28]. Our findings provide further support for this hypothesis.

Furthermore, in line with previous evidence [29], our findings suggest that empathy towards dogs may prompt motivation for acquisition regardless of the dog’s age. In a study which examined the interaction between victim age and species using fictitious news reports, it was found that age affected participants’ degree of empathy toward human victims, but not for dog victims [29]. This suggests that people may consider dogs as vulnerable, whether young or adult. It also implies that dogs are understood as having many of the same characteristics associated with human babies, which contributes to a belief that dogs are unable to fully protect themselves. Our study lends support to this suggestion, because participants unexpectedly acquired dogs of a variety of ages, frequently citing the dog’s vulnerability. Another previous study demonstrated that the presence of an accessory (e.g. toys, lead, collar, chain or bandana) in photographs of dogs in need of adoption was associated with significantly longer lengths of stay online [30]. The authors of this previous study suggest that this may indicate that some potential adopters are seeking a dog to “save” rather than one who appears to be an owned dog. The findings of this current study suggest that unplanned acquisitions frequently involve dogs perceived as needing to be “saved”.

In addition to an emotional response prompted by a dog’s perceived vulnerability, our study identified the significance of emotional feelings towards a dog more generally in people who acquired a dog without intending to do so. In a previous study, Tesfom and Birch (2013) [6] framed adoption decisions in the context of consumer behaviour literature. The various models for understanding consumer decisions include the cognitive and emotional views of the consumer [31]. The cognitive model depicts consumers as information-seeking and undergoing a process of contemplation as they seek to make satisfactory decisions. Meanwhile, the emotional view of decision-making portrays consumers as associating feelings or emotions with purchases. Within their study, Tesfom and Birch [6] considered dog adoption decision-making to resemble elements of both the cognitive and emotional models. They report that although the majority of dog adoption decisions are well-considered, with pre-acquisition research conducted, some adoptions (e.g., an onlooker adopting a dog hit by a car) are based on emotions. In the present study, emotions were often linked to decisions related to the acquisition of dogs in unplanned circumstances. Such emotional decision-making may help to explain the reported mismatches between participants’ intentions and actual acquisition behaviour that were identified in our study. This finding also supports previous research [21,22] in which immediate or quick emotional connections to pets emerged as a key theme amongst people who had acquired a pet with no intention to do so.

Emotions play an important role in the process of impulse buying [32], which has been defined as instances in which “a consumer experiences a sudden, often powerful and persistent urge to buy something immediately” [33]. For many participants in our study, the decision to acquire a dog without prior intention resembles the behaviour of impulse buying. Previous evidence suggests that such impulsive decision-making could lead to undesired consequences in the future, such as less satisfaction with the dog [2]. For instance, a study found that dog owners who were less consistent in their planned and actual acquisition, potentially due to impulse buying, later perceived their dog as more costly (both emotionally and financially) [2].

Unsuccessful dog–human relationships might be characterised by the occurrence of undesired behaviours, ranging from training-based issues such as poor recall to aggressive behaviour. Problem behaviours can lead to mistreatment of the dog, for example, neglect or abuse [34], as well as euthanasia [35] or relinquishment [36]. Such undesired outcomes have been associated with a mismatch between dog-related features, for example, size, age, breed, health, and behaviour, the owner’s knowledge and capabilities regarding the dog’s needs, and the owner’s expectations of their relationship with the dog [2]. Significantly, the success of dog–human relationships are understood to be associated with the owner’s decision-making process prior to acquisition [2]. However, other research has found no significant relationship between receiving a pet as a gift and the recipient’s self-reported attachment toward the pet [37]. Only a few owners within our study reported low levels of satisfaction with their dog. However, because this study did not aim to assess the impact of type of acquisition on ownership outcomes, future research to understand how unplanned acquisitions might be associated with ownership outcomes is warranted.

The results of our study highlight an informal mode of rehoming dogs “within the community” that bypasses organisational structures such as rescue centres and other animal welfare organisations. While previous research has estimated that around 130,000 dogs enter U.K. rehoming charities each year [38], our findings suggest that the scope of dogs moving between homes may be much greater than previously suggested. Rehoming dogs “within the community” is different from relinquishing a dog to an animal rescue organisation. For example, due to privacy reasons, follow-up by previous owners regarding the rehoming status of a dog is not typically possible in rescue centres. However, finding a new home for one’s dog amongst family, friends or even the wider community might enable a previous owner to keep informed of the dog’s whereabouts, health and wellbeing, or even maintain a relationship through visits. Indeed, the current study found that, in cases in which dogs were acquired by a person known to the former owner, contact was often maintained between the dog and previous owner. Several previous studies indicate that relinquishment can be an emotionally troubling experience for owners [39,40,41]. In addition, a study of people who had relinquished their pets via Gumtree found that one of the reasons for choosing to rehome their pet in this manner was that they could evaluate potential new owners [42]. Rehoming to a known person might therefore be a strategy used to minimise negative emotional effects associated with relinquishment by maintaining some control over the process. The aforementioned study found that owners also used Gumtree to rehome their pets because of a perception that animal shelters were full [42]. Although our study did not capture perspectives from previous (i.e., relinquishing) owners, our findings did identify would-be owners’ concerns about alternatives for the dog, including potential relinquishment to a rescue centre, as a reason driving people to acquire a dog unexpectedly.

## 5. Strengths and Limitations

This study is one of the few, and to the authors’ knowledge the most extensive so far, qualitative research studies that explores how and why people acquire dogs. However, because our study used a convenience sample, it may suffer from sampling bias. Participants were recruited whilst they attended Dogs Trust community events. Attending these events aimed to support dog welfare is arguably a behaviour indicative of at least some degree of owner responsibility. The study’s reliance on owners attending these events may introduce a potential limitation regarding the absence of any accounts of the experiences of the “harder to reach” population of dog owners who choose not to engage with such events. Additionally, due to the nature of the convenience sample used, experiences of people with less successful relationships with dogs acquired in an unplanned manner may be absent from our study. Finally, because our study focused on the experiences dog owners in areas of the United Kingdom where there may be a higher prevalence of stray dogs, and/or areas of deprivation, the findings might not reflect experiences in other parts of the country. Thus, future research should be undertaken with more representative samples of dog owners to determine how widespread the phenomenon of unplanned acquisitions is and to identify risk factors for this type of acquisition.

## 6. Conclusions

This study found that many owners had acquired their dog without prior intention to do so. Participants attributed their unplanned acquisitions to their perceptions of the dog’s vulnerability and the quick emotional attachments they formed with the dog. Due to the qualitative nature of this research, conclusions should not be drawn about the actual incidence of unplanned acquisitions in the U.K. population. Future work should explore the phenomenon of unplanned acquisitions within larger populations of dog owners to better clarify and quantify this type of acquisition. One important area for future study is the long-term outcome of dog–human relationships, both when dogs are acquired unexpectedly and when they are rehomed within the community. In addition, further research involving those seeking to rehome their dogs through informal routes is required to fully understand the motivations for this phenomenon. This information can then lead to the design of effective interventions to promote responsible dog acquisition, ownership, and rehoming behaviour to improve dog welfare and dog–human relationships.

## Data Availability

The data presented in this study are available on request from the corresponding author. The data are not publicly available due to the personal and subjective nature of the data.

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
