# Peer review of "“Don’t Bring Me a Dog…I’ll Just Keep It”: Understanding Unplanned Dog Acquisitions Amongst a Sample of Dog Owners Attending Canine Health and Welfare Community Events in the United Kingdom"

_animals, 2021, doi:10.3390/ani11030605_

Round 1
Reviewer 1 Report
The authors should be commended on the completion of such important research. It has been conducted in a scientifically sound way using qualitative methods, which are commonly neglected despite the valuable contribution they make. Please see specific comments below:
L41: It would be valuable to include a few sentences on why understanding acquisition methods may be important, be it a potential influencer of the human-dog relationship or an influencer of relationship outcome (i.e. relinquishment/retention).
L51: Both examples provided are demographic, including one that is classified as socioeconomic would be appropriate.
L81-83: It would be beneficial to include a citation to support your claim of the widespread assumption of preplanned acquisition by researchers.
L127: It is worth considering including, either as a table or in your appendix, the interview schedule implemented in this study. This will provide context for the reader.
L451: Rectify heading format
L645: It is also worth mentioning that the events were held in areas perceived to be more vulnerable, with higher relinquishment rates/stray dogs. These findings may not accurately reflect the situation elsewhere.
Reviewer 2 Report
Thank you for providing me with the opportunity to review this paper. I thought it was very interesting and thorough.
Abstract
Line 35: ‘more complete accurate’ seems to be missing a comma or ‘and’.
Introduction
Lines 80-82: please provide references to support the notion that interventions and research have assumed acquisition is a pre-planned process.
Methods
Could you please provide the semi-structured interview guide, perhaps as supplementary material?
Were there any changes made to the interview guide because of the pilot testing?
Results
Please also provide the exact number of participants who reported an unplanned acquisition and are included in the results.
It would also be useful for the reader if the authors could provide exact numbers/percentages of how common each acquisition type was, i.e., family/friends, gift etc.
Line 451: Is ‘Inconsistency Between Plans for a Future Dog and the Actual (Unplanned Acquisition)’ meant to be a sub-heading?
Line 503: How many participants reported challenges with ownership?
Discussion
In the paragraph starting at line 601, the authors describe a possible association between impulsive dog acquisition and lower ownership satisfaction, but their results do not support this hypothesis given most participants had a positive relationship with their dog (line 496). There has been other research that has shown unplanned acquisitions, such as pets given as gifts, are not associated with lower attachment levels or an increased likelihood of relinquishment. The following paragraph also implies that owners with unplanned acquisitions may experience difficulties with problem behaviors. It seems that few owners in the current study reported challenges with undesirable behavior with the exception of one owner who acquired a dog with known behavioral problems. I certainly agree that future research investigating the effect of unplanned acquisitions on the human-dog relationship is needed. But, unless there is evidence to the contrary, I think it’s important to acknowledge that the participants in this study seem to have strong levels of attachment to their dogs and that this type of acquisition is not inherently bad.
Reviewer 3 Report
This is a well-researched, well-written paper that has practical implications for animal rehoming. You might consider Irvine's book, If You Tame Me, to see her examination of unplanned adoptions in an animal shelter.
